# Prevalence of Extended-Spectrum *β*-Lactamase-Resistant Genes in *Escherichia coli* Isolates from Central China during 2016–2019

**DOI:** 10.3390/ani12223191

**Published:** 2022-11-18

**Authors:** Zui Wang, Qin Lu, Xiaohui Mao, Li Li, Junfeng Dou, Qigai He, Huabin Shao, Qingping Luo

**Affiliations:** 1Key Laboratory of Prevention and Control Agents for Animal Bacteriosis (Ministry of Agriculture and Rural Affairs), Institute of Animal Husbandry and Veterinary, Hubei Academy of Agricultural Sciences, Wuhan 430064, China; 2Hubei Hongshan Laboratory, Wuhan 430064, China; 3State Key Laboratory of Agricultural Microbiology, College of Veterinary Medicine, Huazhong Agricultural University, Wuhan 430064, China

**Keywords:** cephalosporin, *Escherichia coli*, extended-spectrum *β*-lactamase, multiplex qPCR, resistance gene

## Abstract

**Simple Summary:**

*β*-lactam antibiotics are commonly used for the treatment of severe infection in both animals and humans, and the resistance of *E. coli* to third-generation cephalosporins is becoming a worldwide problem. Our results revealed 407 *E. coli* strains isolated from central China exhibited strong resistance to first- to fourth-generation cephalosporins and monobactam antibiotics, and piperacillin/tazobactam was the most effective drug. Phenotypically, 63.88% of the isolates were positive for ESBL production and the isolation rates kept growing. Genetic characterization identified CTX-M as the most prevalent type and *bla*_TEM_ + *bla*_CTX-M_ as the most common ESBL genotype combination. Furthermore, a novel multiplex real-time PCR method for detecting the three most common ESBL genes was developed as a complementary rapid-screening test for antimicrobial resistance genes. These results confirm the cephalosporins resistance urgency in the central Chinese poultry sector and suggest the continuous monitoring and timely detection of ESBL-producing isolates is vitally important in establishing appropriate antimicrobial therapies and blocking their transmission.

**Abstract:**

The emergence and dissemination of *Escherichia coli* (*E. coli*) strains that produce extended-spectrum beta-lactamases (ESBLs) represents a major public health threat. The present study was designed to evaluate the prevalence and characteristics of ESBL-producing *Escherichia coli* isolates from chickens in central China during 2016–2019. A total of 407 *E. coli* strains isolated from 581 chicken swabs were identified conventionally and analyzed for various cephalosporin susceptibility by disk-diffusion assay. ESBL-producing strains were screened using the double=disk synergy test and ESBL-encoding genes were carried out by PCR/sequencing. A total of 402 *E. coli* isolates exhibited strong resistance to first- to fourth-generation cephalosporins and monobactam antibiotics, especially cefazolin (60.69%), cefuroxime (54.05%), cefepime (35.14%), ceftriaxone (54.30%), and aztreonam (40.29%). Piperacillin/tazobactam (1.72%) was the most effective drug against the strains, but the resistance rates increased each year. Among the isolates, 262 were identified as ESBL producers and the isolation rates for the ESBL producers increased from 63.37% to 67.35% over the four years. CTX-M (97.33%) was the most prevalent type, followed by TEM (76.72%) and SHV (3.05%). The most common ESBL genotype combination was *bla*_TEM_ + *bla*_CTX-M_ (74.46%), in which the frequency of carriers increased steadily, followed by *bla*_CTX-M_ + *bla*_SHV_ (3.05%). In addition, the most predominant specific CTX-M subtypes were CTX-M-55 (48.47%) and CTX-M-1 (17.94%), followed by CTX-M-14 (11.01%), CTX-M-15 (8.02%), CTX-M-9 (6.11%), CTX-M-65 (4.58%), and CTX-M-3 (1.15%). Moreover, a novel multiplex qPCR assay was developed to detect *bla*_CTX-M_, *bla*_TEM_, and *bla*_SHV_, with limits of detection of 2.06 × 10^1^ copies/μL, 1.10 × 10^1^ copies/μL, and 1.86 × 10^1^ copies/μL, respectively, and no cross-reactivity with other ESBL genes and avian pathogens. The assays exhibited 100% sensitivity and specificities of 85%, 100%, and 100% for *bla*_CTX-M_, *bla*_TEM_, and *bla*_SHV_, respectively. In conclusion, our findings indicated that ESBL-producing *E.coli* strains isolated from chickens in central China were highly resistant to cephalosporins and frequently harbored diversity in ESBL-encoding genes. These isolates can pose a significant public health risk. The novel multiplex qPCR method developed in this study may be a useful tool for molecular epidemiology and surveillance studies of ESBL genes.

## 1. Introduction

*Escherichia coli* (*E. coli*) is one of the main pathogens that cause diarrhea in pigs and chickens, and it is an important cause of death among piglets and young poultry [1]. Antibiotics are generally used to treat *E. coli* infections due to the lack of an effective vaccine. However, bacteria with resistance to multiple drugs have increasingly been found because of the overuse and misuse of antibiotics on farms to reduce and prevent the risk of infection [2]. Pigs and chickens are food-producing animals but are also major reservoirs of organisms with antimicrobial resistance [3,4]. In addition to antibiotic residues in food, antibiotic resistance genes can disseminate between animals and humans via the food chain [5]. Therefore, drug-resistant *E. coli* strains pose severe hazards to animal husbandry and food safety.

Resistance to third-generation cephalosporins is growing, so much attention has been increasingly focused on extended-spectrum *β*-lactamase (ESBL) enzymes that can hydrolyze the *β*-lactam ring in a wide range of *β*-lactam antimicrobials [6]. ESBL-producing *E. coli* isolates are also among the most critical antibiotic-resistant pathogens categorized by the World Health Organization. In general, the most prevalent ESBLs are encoded by *bla*_TEM_, *bla*_CTX-M_, and *bla*_SHV_, and the frequency of carrying *bla*_CTX-M_ is highest in ESBL-producing *E. coli*. Up to 97.8% of *E. coli* isolates from pigs, cattle, chicken, and sheep produce CTX-M group enzymes [7]. In recent years, the extensive use of antimicrobial agents, coupled with the transmissibility of resistance determinants, such as integrons, has resulted in the emergence and spread of MDR ESBL-producing strains. Previous studies have demonstrated that E. coli is one of the major causes of childhood diarrhea, and the distribution of class 1 integrons and ESBLs in the strains is highly prevalent [8]. The production of ESBLs can greatly restrict the uses of clinical medication, so the continuous monitoring and timely detection of ESBL-producing isolates is vitally important in establishing appropriate antimicrobial therapies and blocking their transmission [9]. At present, the common methods used to identify ESBL-producing bacteria include the drug-sensitive disk-diffusion method, microbroth-dilution method, and E-test method. However, these are culture-based methods for phenotypic screening, which are time-consuming, laborious, and incapable of distinguishing different ESBL genotypes. Molecular techniques, such as multiplex PCR and real-time PCR, have been applied to detect the specific genes that encode ESBLs [10,11]. The emergence of ESBL enzymes is mainly due to plasmids and chromosomal mutations [12]. Unfortunately, many new mutations have been found because of the misuse of third-generation cephalosporins, thereby leading to the emergence of new drug-resistant gene subtypes. For example, the CTX-M genotypes include CTX-M-1, CTX-M-2, CTX-M-8, CTX-M-9, and CTX-M-25 [13]. Therefore, it is necessary to improve the existing molecular detection techniques. In the present study, we developed a novel multiplex real-time PCR method to detect the three most common ESBL genes (*bla*_TEM_, *bla*_CTX-M_, and *bla*_SHV_) and to identify the trends in cephalosporin resistance among *E. coli* isolates obtained from chicken feces samples collected in central China during 2016–2019.

## 2. Methods

### 2.1. Collection of Bacterial Strains

From 2016 to 2019, a total of 581 swab samples were obtained from broiler chickens at different farms located in central China, including Wuhan, Xiangyang, Xiantao, Xinyang, Daye, Tianmen, Jinan, Hanchuan, Changde, and Yucheng. The collected samples were streaked onto MacConkey agar and eosin-methylene blue agar (Haibo, Qingdao, China), and then incubated at 37 °C for 24 h. Typical *E. coli* colonies were subsequently confirmed by PCR amplification of the *phoA* gene using *E. coli*-specific primers. In addition, reference strain ATCC 29522 was obtained from the China Veterinary Culture Collection Center. Unless otherwise stated, all cultures were maintained in Luria broth (Haibo) in the presence of oxygen at 37 °C.

### 2.2. Antimicrobial Susceptibility Testing

The susceptibility of *E. coli* isolates to antimicrobial agents was determined using the disk-diffusion assay according to the methods of the Clinical and Laboratory Standards Institute (CLSI). Briefly, antimicrobial disks were placed on the surfaces of Mueller–Hinton agar (Haibo) plates inoculated with *E. coli*. The inoculated plates were incubated at 37 °C for 24 h, before the diameters of the zones of inhibition were measured and interpreted following CLSI guidelines. The *E. coli* ATCC 25922 strain was used for quality control in the test. The antimicrobial disks (Oxoid, England, UK) contained cefazolin (CZO), cefuroxime (CXM), cefoxitin (FOX), ceftriaxone (CRO), ceftazidime (CAZ), cefoperazone/sulbactam (SCF), cefepime (FEP), aztreonam (ATM), amoxicillin/clavulanic acid (AMC), ampicillin/sulbactam (SAM), and piperacillin/tazobactam (TZP).

### 2.3. Screening for ESBL-Producing Strains

The *E. coli* isolates were phenotypically screened using the double-disk synergy test. Briefly, antimicrobial disks containing CAZ (30 μg) and CAZ/clavulanic acid (30/10 μg), or cefotaxime (30 μg) and cefotaxime/clavulanic acid (30/10 μg) (Solarbio, Beijing, China), were placed on the surfaces of Mueller–Hinton agar plates inoculated with *E. coli*. After inverted culture at 37 °C for 24 h, the diameters of the zones of inhibition were measured. ESBL production was indicated when the zone of inhibition in the CAZ or cefotaxime group was expanded by clavulanate (≥5 mm). *E. coli* ATCC 25922 was used as the negative control.

### 2.4. Characterization of ESBL Genes by PCR Sequencing

The ESBL genes (*bla*_TEM_, *bla*_CTX-M_, and *bla*_SHV_) were detected in ESBL-producing *E. coli* isolated by PCR and confirmed by DNA sequencing. Briefly, nucleic acid was extracted from the isolates using the boiling method. PCR was performed using gene-specific primers (Table 1) and the products were sequenced by Sangon Biotech (Shanghai, China).

### 2.5. Probe and Primer Design

Genetic sequences of *bla*_TEM_, *bla*_CTX-M_, and *bla*_SHV_ were downloaded from the GenBank database and aligned using Megalign (Madison, WI, USA). The primers and probes were designed using Primer Express 3.0 (ABI, La Jolla, CA, USA) software based on the conserved sequences (Table 1). Subsequently, the specificities of the primers and probes were evaluated by BLAST searching against the GenBank non-redundant database. Appropriate fluorophore groups and quenched groups were labeled at the 5′ and 3′ ends of the probes (Table 1). All primers were synthesized by Sangon Biotech.

### 2.6. Multiplex Quantitative PCR Amplification

The optimal qPCR system contained 5 μL of AceQ^®^ qPCR Probe Master Mix (Vazyme, Nanjing, China), 0.4 μM of primers (TEM), 0.2 μM of primers (CTX-M), 0.3 μM of primers (SHV), 0.8 μM of primers (TEM), 0.4 μM of primers (CTX-M and SHV), 1 μL of cDNA, and 1.5 μL of nuclease-free H_2_O. The optimum reaction temperature was tested from 55.6 °C to 64.6 °C, with a temperature gradient of 1.5 °C. The specificity of the qPCR assay was examined with *E. coli* strains, including ESBL-producing strains and *E. coli* ATCC 25922. Nuclease-free H_2_O was used for the negative controls. Amplification and signal detection were conducted with a Bio-Rad system (Bio-Rad, Hercules, CA, USA).

### 2.7. Sensitivity Tests and Standard Curve

The detection targets were cloned into the pMD18-T vector and the concentrations of pMD-CTX-M, pMD-TEM, and pMD-SHV were determined using a NanoDrop One/One^C^ spectrophotometer (Thermo, Waltham, MA, USA). Sensitivity and standard curve tests were conducted using a 10-fold dilution series of the recombinant plasmids. Nuclease-free H_2_O was used as the negative control. Sensitivity tests were performed under the optimized reaction conditions and the results were compared to the PCR results. Standard curves were also established.

### 2.8. Detection of ESBL Genes Using the Novel Multiplex qPCR Method

All of the ESBL-positive *E. coli* isolates were screened for the presence of *bla*_TEM_, *bla*_CTX-M_, and *bla*_SHV_ using the novel multiplex qPCR method. The results obtained by multiplex qPCR were compared to those produced using the normal PCR method. The agreement between the two methods was measured based on the significance using kappa (κ) statistics in QuickCals software (https://www.graphpad.com/quickcalcs/kappa1 accessed on 5 May 2022) (Montrose, CA, USA). The sensitivity and specificity were measured with MedCalc software (https://www.medcalc.org/calc/diagnostic_test.php accessed on 5 May 2022) (Ostend, Belgium).

## 3. Results

### 3.1. Antimicrobial Susceptibility and Phenotypic Identification

During 2016–2019, 407 *E. coli* strains were obtained using cloacae swabs from chickens in central China and 64.37% (262/407) were ESBL producers (Table 2), with 63.37% (64/101) in 2016, 60.42% (58/96) in 2017, 66.07% (74/112) in 2018, and 67.35% (66/98) in 2019. The isolation rate of ESBL producers increased each year, except in 2017.

Among the 407 isolates tested with different antibiotics, the resistance rates were as follows: CZO = 247/407 (60.69%); CXM = 220/407 (54.05%); FEP = 143/407 (35.14%); CRO = 221/407 (54.30%); ATM = 164/407 (40.29%); SAM = 91/407 (22.36%); FOX = 64/407 (15.72%); CAZ = 57/407 (14.05%); SCF = 25/407 (6.14%); AMC = 51/407 (12.53%); and TZP = 7/407 (1.72%). TZP was the most potent antibiotic with a susceptibility of 98.28%, followed by SCF (93.86%), but the resistance rates to TZP and SCF tended to increase during the four years (Figure 1). Similar to the phenotypic test results, the resistance rates to the major antibiotics were lower in 2017. In addition, all of the ESBL producers (100%, 262/262) were resistant to at least one or more *β*-lactam antibiotic, and the resistance rate was 16.55% (24/145) in non-ESBL producers, which increased rapidly (Table 2).

### 3.2. Genotypic Characterization of ESBL Producers

Among the 262 ESBL-positive isolates, all isolates were positive for ESBL genes, thereby indicating the high sensitivity of the phenotypic tests (Table 3). Among the ESBL-positive isolates, *bla*_CTX-M_ genes were detected in 97.33% (255/262) of the isolates and *bla*_TEM_ genes were detected in 76.72% (201/262) of the isolates. However, *bla*_SHV_ genes had the lowest detection rate of (3.05%) 8/262, and they were only detected in 2019. In addition, the most common ESBL genotype combination in *E. coli* was *bla*_TEM_ + *bla*_CTX-M_ (194/262, 74.46%), and the rate increased steadily (Table 3). By contrast, the detection rates for *bla*_TEM_ (7/262, 2.67%) and *bla*_CTX-M_ (53/262, 20.23%) alone decreased slowly. Furthermore, the CTX-M genotypes include several variants (CTX-M-1, CTX-M-2, CTX-M-8, CTX-M-9, and CTX-M-25) but only two CTX-M genotypes (CTX-M-1G, 75.57%; CTX-M-9G, 21.76%) were detected in the isolates (Figure 2). The predominant specific CTX-M types were CTX-M-55 (48.47%) and CTX-M-1 (17.94%), followed by CTX-M-14 (11.01%), CTX-M-15 (8.02%), CTX-M-9 (6.11%), CTX-M-65 (4.58%), and CTX-M-3 (1.15%). It should be noted that the rate of CTX-M-14, CTX-M-15, CTX-M-65, and CTX-M-3 detection increased each year. Moreover, some new combinations of ESBL genotypes were developed in the last two years, including CTX-M-3/TEM-1, CTX-M-15/TEM-30, CTX-M-15/SHV-12, CTX-M-55/SHV-12, and CTX-M-65/SHV-12.

### 3.3. Establishment of a Novel Multiplex qPCR Method

Primers and probes were designed based on the conserved sequences in order to establish a novel multiplex qPCR method to detect the three common ESBL genes (*bla*_TEM_, *bla*_CTX-M_, and *bla*_SHV_) (Figure 3). Some point mutations cannot be avoided when detecting the targets in the CTX-M groups, so primers and probes were inserted with degenerate bases. The optimal reaction conditions for multiplex qPCR comprised initial denaturation at 95 °C for 30 s, followed by 40 cycles at 95 °C for 5 s and 57 °C for 30 s.

In the qPCR specificity tests, only ESBL-producing strains produced three positive fluorescence signals, and no positive signals were observed with ATCC 29522 and nuclease-free H_2_O (Figure 4A).

In the qPCR sensitivity tests, the detection limits for *bla*_CTX-M_, *bla*_TEM_, and *bla*_SHV_ were determined as 2.06 × 10^1^ copies/μL, 1.10 × 10^1^ copies/μL, and 1.86 × 10^1^ copies/μL in a single-tube assay system, respectively, and the limits were 2.06 × 10^3^ copies/μL, 1.10 × 10^3^ copies/μL, and 1.86 × 10^4^ copies/μL in the PCR sensitivity tests (Figure 5). Thus, the sensitivity of the new method was at least 100 times that of the PCR assays. Using the standard curves established for the three ESBL genes (Figure 4B), three linear relationships were obtained between the copy number (x-axis) and quantification cycle (Cq) value (y axis) as follows: y = −3.328x + 38.561, R^2^ = 0.999 for *bla*_TEM_; y = −3.309x + 38.09, R^2^ = 0.998 for *bla*_CTX-M_; and y = −3.304x + 37.928, R^2^ = 0.999 for *bla*_SHV_.

### 3.4. Comparison of the Novel Multiplex qPCR Method and PCR

To determine the efficiency of the novel multiplex qPCR method, 262 ESBL-producing *E. coli* isolates were analyzed by multiplex qPCR and normal PCR. The results are compared in Table 4.

Among the 262 strains tested, 53 (20.23%) were positive for the presence of *bla*_CTX-M_ alone and eight (3.05%) were positive for the presence of the *bla*_CTX-M_ + *bla*_SHV_ combination using multiplex qPCR. The same results were obtained with the PCR method, which is considered the gold standard. Six strains (6/262, 2.29%) were detected as positive for *bla*_TEM_ alone by multiplex qPCR and seven strains (7/262, 2.67%) were detected as positive by PCR. The strain with a different result was detected as the combination of *bla*_CTX-M_ + *bla*_TEM_ using multiplex qPCR, thereby resulting in the discrepancy for one of the *bla*_CTX-M_ + *bla*_TEM_ samples. Both assays were negative for the detection of other *β*-lactamase gene patterns. Therefore, the novel multiplex qPCR method obtained almost perfect agreement with the gold standard PCR method (*κ* = 0.921 for *bla*_CTX-M_ and *κ* = 1.000 for both *bla*_TEM_ and *bla*_SHV_). In addition, the specificity, sensitivity, positive predictive value (PPV), and negative predictive value (NPV) results are shown in Table 5.

## 4. Discussion

*E. coli* infection remains a major health concern in poultry because it can be transmitted to humans through the food chain [14]. *E. coli* infections severely threaten the health of humans and poultry, as well as the development of the poultry industry and the economy. The resistance of *E. coli* to third-generation cephalosporins is a worldwide problem [2], mainly caused by the production of ESBLs. The transfer of ESBLs between bacteria is usually mediated by plasmids [15]. More than 600 ESBLs have been reported [16], and the majority are TEM, SHV, and CTX-M-type *β*-lactamases. A previous study showed that poultry feces could play an important role as potential reservoirs for resistant isolates and resistance genes [17]. In the present study, 407 avian *E. coli* strains were isolated from fecal samples collected during 2016–2019 in central China. The isolates exhibited severe resistance to first- to fourth-generation cephalosporins and monobactam antibiotics, especially CZO (60.69%), CXM (54.05%), FEP (35.14%), CRO (54.30%), and ATM (40.29%). The resistance rates were much higher than those found in other areas of China [18,19,20]. It indicated that the *E. coli* recovered from chickens developed serious resistance to cephalosporins; therefore, the use of these drugs should be suspended. TZP is effective against ESBL-producing Gram-negative bacteria and it is used as a first-line drug to treat serious infections [21]. In our study, TZP was the most potent antibiotic with a susceptibility of 98.28%. Although it could be used to treat the resistant isolates, the drug resistance rates increased each year, which reminds people to use antibiotics correctly and appropriately. ESBLs can hydrolyze all types of cephalosporins and monobactams, so they are the most significant in the spread of resistant determinants worldwide, mainly in *E. coli* [22]. The current study identified a prevalence (64.37%, 262/407) of ESBL producers. The isolation rates of ESBL producers increased from 63.37% to 67.35% during the four years, and the rates were much higher than those found in previous similar studies of chickens in China during 2004–2007 (28.8%) [23] and 2008–2014 (41.53%) [24]. Among the ESBL-positive isolates, the dominant resistance gene was *bla*_CTX-M_ (97.33%), and the most common genotype combination was *bla*_TEM_ + *bla*_CTX-M_ (74.46%). Compared with the OXA and SHV types, the CTX-M and TEM types have spread more widely and they are now dominant in China, possibly because cefotaxime and CRO are the most commonly used clinical drugs for the treatment of bacterial diseases caused by *Enterobacterales* [25,26]. In addition, the CTX-M group is used most widely in China, so CTX-M-1G (75.57%) and CTX-M-9G (21.76%) were detected in ESBL-producing strains, and the detection rate of CTX-M-9G increased each year. In some regions of Asia, CTX-M-9G is the predominant group [27] and CTX-M-9-producing *E. coli* are always detected in urinary tract infections [28]. Moreover, previous studies have shown that two or more CTX-M groups frequently coexist in the same isolate, which could promote the occurrence of other recombinant enzymes in the future [29]. Fortunately, CTX-M-1 and CTX-M-9 group members were not found to coexist in the isolates in the present study. Additionally, the predominant specific CTX-M types were CTX-M-55 (48.47%) and CTX-M-1 (17.94%), followed by CTX-M-14 (11.01%), CTX-M-15 (8.02%), CTX-M-9 (6.11%), CTX-M-65 (4.58%), and CTX-M-3 (1.15%). A high carrying rate of CTX-M-55 has been reported in *E. coli* previously in many countries, including Hong Kong and Vietnam [30,31]. The emergence of CTX-M-55 has also been detected in *Salmonella* strains from raw meat and food animals in China [32]. Moreover, many avian ESBL *E. coli* isolates can produce CTX-M-14, CTX-M-15, and CTX-M-65 commonly detected in clinical samples, which is worrying in terms of food safety. CTX-M-15 has become the most prevalent CTX-M variant in both hospitals and the environment worldwide [33]. CTX-M-3 is frequently found in *Salmonella* and rarely detected in *E. coli*. However, it should be noted that the rate of CTX-M-14, CTX-M-15, CTX-M-65, and CTX-M-3 detection increased each year in the present study. On the other hand, the phenotypic resistance to cephalosporins was not validated completely by the genotypic ESBL results, possibly due to the presence on another resistance mechanism, such as AmpC *β*-lactamases [34,35], which was one of the reasons for the reduction in sensitivity towards newer generations of antibiotics [36]. In addition, ESBL-producing strains with AmpC *β*-lactamases could cause a false-negative ESBL production [8]. In summary, our results showed that the isolates exhibited very high resistance to cephalosporins and the frequency of ESBL producers increased during 2016–2019. Therefore, the early detection and characterization of ESBLs are vitally important in establishing appropriate antimicrobial therapies and preventing their transmission.

The routine diagnostic methods currently used to detect ESBL producers involve culture-based phenotypic screening, which is time-consuming and laborious, and they cannot distinguish different ESBL genotypes [37]. Molecular detection techniques have potential advantages due to the limitations of phenotypic detection methods. In the present study, we developed a novel multiplex qPCR assay for the rapid detection of *bla*_CTX-M_, *bla*_TEM_, and *bla*_SHV_ with high specificity and low detection limits. CTX-M genotypes comprise CTX-M-1, CTX-M-2, CTX-M-8, CTX-M-9, and CTX-M-25 with a low homology and a few conserved regions among each group. Therefore, it is difficult to design probes and primers to detect all of these genotypes at the same time. CTX-M-1 and CTX-M-9 were the predominant types found in avian *E. coli* isolates in China and we only detected these genotypes in the 262 ESBL producers by PCR in the present study. Thus, the primers used to detect CTX-M could detect CTX-M-1 and CTX-M-9 at the same time. Some point mutations cannot be avoided when detecting targets in CTX-M groups, so the primers and probe were inserted with degenerate bases. The detection limits in the multiplex qPCR assays in the single-tube assay system were determined as 2.06 × 10^1^ copies/μL for *bla*_CTX-M_, 1.10 × 10^1^ copies/μL for *bla*_TEM_, and 1.86 × 10^1^ copies/μL for *bla*_SHV_, and they were around 100 to 1000 times higher than those using the conventional PCR method. In addition, our assays were conducted using various resistance genes and ATCC 29522. Only ESBL-producing strains produced three positive fluorescence signals, and no positive signals were obtained with ATCC 29522 and nuclease-free H_2_O. In the efficiency tests, 262 ESBLs producers were detected using the normal PCR method and multiplex qPCR assay. The qPCR assay results were in almost perfect agreement with the PCR results, except for one sample. Therefore, the results demonstrated that the novel multiplex qPCR assay was highly efficient at detection with high specificity; thus, it can be used as a clinical detection method or as a complementary rapid-screening test for antimicrobial-resistant genes while phenotypic tests are being conducted.

## 5. Conclusions

In the present study, ESBL-producing *E. coli* isolates from chickens in central China during 2016–2019 exhibited high resistance to cephalosporins and they frequently carried diverse ESBL genes, thereby indicating that chickens have become reservoirs of antimicrobial-resistant organisms and that the continuous monitoring and timely detection of ESBL-producing isolates is vitally important. The novel multiplex qPCR method developed in this study could be effective in monitoring the prevalence of ESBL-producing *E. coli* in chickens to reduce the risk of transmission to humans. 

## Figures and Tables

**Figure 1 animals-12-03191-f001:**
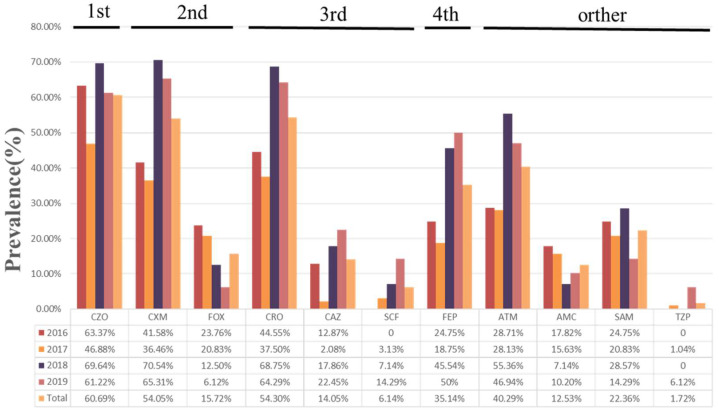
Antimicrobial resistance patterns in 407 *E. coli* isolates. First-generation cephalosporin: CZO, cefazolin. Second-generation cephalosporins: CXM, cefuroxime; FOX, cefoxitin. Third-generation cephalosporins: CRO, ceftriaxone; CAZ, ceftazidime; SCF, cefoperazone/sulbactam. Fourth-generation cephalosporin: FEP, cefepime. Other β-lactam antibiotics: ATM, aztreonam; AMC, amoxicillin/clavulanic acid; SAM, ampicillin/sulbactam; TZP, piperacillin/tazobactam.

**Figure 2 animals-12-03191-f002:**
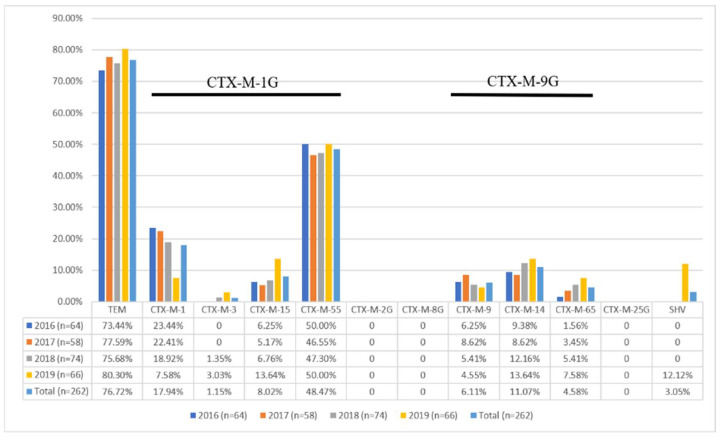
Percentage of ESBL subtypes in ESBL-producing *E. coli* isolates from chicken feces samples (n = 262).

**Figure 3 animals-12-03191-f003:**
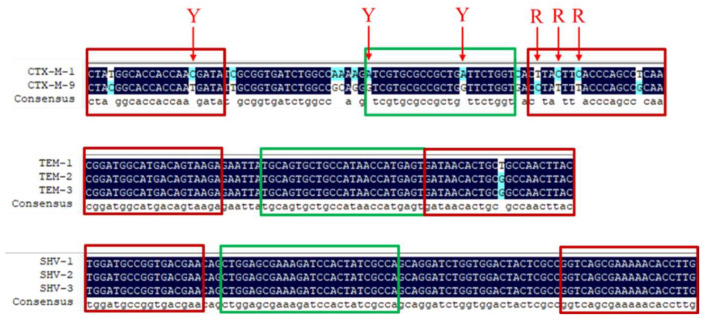
Design of probes and primers for detecting *bla*_CTX-M_, *bla*_SHV_, and *bla*_TEM_ genes.

**Figure 4 animals-12-03191-f004:**
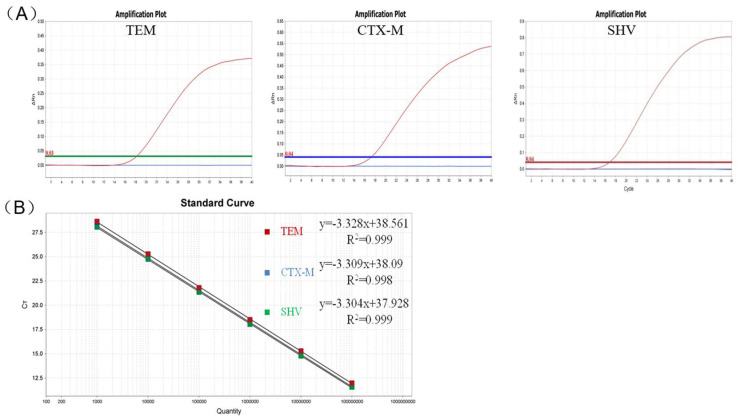
Specificity (**A**) and standard curve (**B**) for multiple qPCR assay of *bla*_TEM_, *bla*_CTX-M_, and *bla*_SHV_ genes.

**Figure 5 animals-12-03191-f005:**
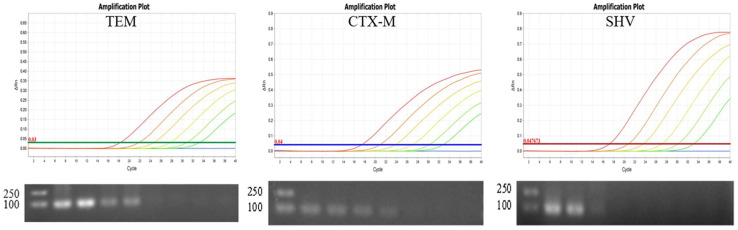
Sensitivity of multiple qPCR assay and PCR for *bla*_TEM_, *bla*_CTX-M_, and *bla*_SHV_ genes.

**Table 1 animals-12-03191-t001:** Primer sequences used for detecting *bla*_CTX-M_, *bla*_SHV_, and *bla*_TEM_ genes.

Primer and Probe	Sequence (5′-3′)	Product Size (bp)
qTEM-F	CGGATGGCATGACAGTAAGA	
qTEM-R	GTAAGTTGGCAGCAGTGTTATC	101
qTEM-P	Hex-TGCAGTGCTGCCATAACCATGAGT-BHQ1	
qCTX-M-F	CTATGGCACCACCAAYGATA	
qCTX-M-R	TTGAGGCTGGGTRAARTARG	86
qCTX-M-P	TAMRA-ACCAGAAYCAGCGGCGCACGAY-BHQ2	
qSHV-F	TGGATGCCGGTGACGAA	
qSHV-R	CAAGGTGTTTTTCGCTGACC	90
qSHV-P	FAM-CTGGAGCGAAAGATCCACTATCGCCA-BHQ1	
TEM-F	AAACGCTGGTGAAAGTA	822
TEM-R	AGCGATCTGTCTAT	
CTX-M-1-F	GGTTAAAAAATCACTGCGTC	850
CTX-M-1-R	TTGGTGACGATTTTAGCCGC	
CTX-M-9-F	ATGGTGACAAAGAGAGTGCA	850
CTX-M-9-R	CCCTTCGGCGATGATTCTC	
CTX-M-2-F	CGACGCTACCCCTGCTATT	552
CTX-M-2-R	CCAGCGTCAGATTTTTCAGG	
CTX-M-8-F	TCGCGTTAAGCGGATGATGC	666
CTX-M-8-R	AACCCACGATGTGGGTAG	
CTX-M-25-F	TTGTTGAGTCAGCGGGTTGA	474
CTX-M-25-R	GCGCGACCTTCCGGCCAAAT	
SHV-F	ATGCGTTATATTCGCCTGTG	753
SHV-R	TGCTTTGTTATTCGGGCCAA	

**Table 2 animals-12-03191-t002:** Antimicrobial susceptibility and phenotypic identification.

Years	Number	ESBL Producers (%)	*β*-Lactam Antibiotics Resistant (%)	
			ESBL Producers	None-ESBL Producers
2016	101	64 (63.37%)	64/64 (100%)	4/37 (10.81%)
2017	96	58 (60.42%)	58/58 (100%)	4/38 (10.52%)
2018	112	74 (66.07%)	74/74 (100%)	8/38 (21.05%)
2019	98	66 (67.35%)	66/66 (100%)	8/32 (25%)
Total	407	262 (64.37%)	262/262 (100%)	24/145 (16.55%)

**Table 3 animals-12-03191-t003:** Frequency and combinations of ESBL genes among in ESBL-producing *E. coli* (n = 262).

Beta-Lactamase Genes in *E. coli*		2016 (n = 64)	2017 (n = 58)	2018 (n = 74)	2019 (n = 66)	Total (n = 262)
TEM-1		2	3	2	-	7
CTX-M		17	13	18	5	53
	CTX-M-1	3	2	5	-	10
	CTX-M-15	2	-	1	-	3
	CTX-M-55	7	8	4	3	22
	CTX-M-9	3	2	1	-	6
	CTX-M-14	2	1	5	1	9
	CTX-M-65	-	-	2	1	3
CTX-M+TEM		45	42	54	53	194
	CTX-M-1, TEM-1	12	11	9	5	37
	CTX-M-3, TEM-1	-	-	1	2	3
	CTX-M-15, TEM-1	2	3	3	5	13
	CTX-M-15, TEM-30	-	-	1	1	2
	CTX-M-55, TEM-1	23	18	27	23	91
	CTX-M-55, TEM-30	2	1	4	3	10
	CTX-M-9, TEM-1	1	3	3	3	10
	CTX-M-14, TEM-1	4	4	4	8	20
	CTX-M-65, TEM-1	1	2	2	3	8
CTX-M+SHV		-	-	-	8	8
	CTX-M-15, SHV-12	-	-	-	3	3
	CTX-M-55, SHV-12	-	-	-	4	4
	CTX-M-65, SHV-12	-	-	-	1	1

**Table 4 animals-12-03191-t004:** Pairwise comparison of *bla*_TEM_, *bla*_CTX-M_, and *bla*_SHV_ detection using multiplex qPCR and PCR (n = 262).

*β*-Lactamase Genes	PCR	Multiplex qPCR		
		*bla* _CTX-M_	*bla* _TEM_	*bla* _SHV_
*bla*_CTX-M_ alone	53	53	0	0
*bla*_TEM_ alone	7	0	6	0
*bla*_SHV_ alone	0	0	0	0
*bla*_CTX-M_ + *bla*_TEM_	194	195	195	0
*bla*_CTX-M_ + *bla*_SHV_	8	8	0	8
*bla*_TEM_ + *bla*_SHV_	0	0	0	0
*bla*_CTX-M_ + *bla*_TEM_ + *bla*_SHV_	0	0	0	0

**Table 5 animals-12-03191-t005:** Agreement of multiplex qPCR and PCR at detecting *bla*_TEM_, *bla*_CTX-M_, and *bla*_SHV_ (n = 262).

Multiplex qPCR	PCR		Sensitivity [IC]	Specificity [IC]	PPV ^a^ [IC]	NPV ^b^ [IC]
CTX-M	Positive	Negative				
Positive	255	1	100 [98–100]	85 [42–99]	99 [97–99]	100
Negative	0	6				
TEM	Positive	Negative				
Positive	201	0	100 [98–100]	100 [94–100]	100	100
Negative	0	61				
SHV	Positive	Negative				
Positive	8	0	100 [63–100]	100 [98–100]	100	100
Negative	0	254				

^a^: positive predictive value; ^b^: negative predictive value.

## Data Availability

All the data are presented in this study.

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
