# Peer review of "Prevalence of Extended-Spectrum β-Lactamase-Resistant Genes in Escherichia coli Isolates from Central China during 2016–2019"

_animals, 2022, doi:10.3390/ani12223191_

Round 1
Reviewer 1 Report
- The current study has a significant impact, but it needs a major revision to improve the quality of the manuscript:
1. Please write the scientific names of bacterial pathogens and genes in the correct form all over the manuscript and in the References section (should be italic).
2. The abstract must illustrate the used methods and the most prevalent results (give more hints about methods and results). Besides, rephrase the aim of the work and the main conclusion of your findings.
3. Discuss more on significant results of this study.
4. Please discuss more on alarming results of this research in discussion section.
5. Recheck all numbers and percents in the text, to ensure their accuracy.
6. Some references are not based on the journal guidelines, please re-check all references.
7. Minor spacing problems exist between the words in some parts of the text.
8. The English of the manuscript should be reviewed and syntax and errors should be corrected before publication.
9. Use the following valuable studies performed on pathogen isolates in the introduction or discussion section and add related references, including:
https://doi.org/10.1007/s11033-022-07215-5
https://doi.org/10.1007/s11033-020-06047-5
https://doi.org/10.4103/bbrj.bbrj_270_21
https://doi.org/10.24171/j.phrp.2021.0272
https://doi.org/10.1371/journal.pone.0266787
https://doi.org/10.3923/jbs.2009.820.824
Reviewer 2 Report
This manuscript presents data about extended spectrum b-lactamases (ESBLs) resistance in Escherichia coli isolates from cloacae from chickens during 2016-2019. Moreover, the authors implemented an qPCR method to detect and quantify ESBLs genes as a tool for molecular epidemiology and surveillance studies of ESBLs genes.
The work is interesting and presents important results in epidemiological terms. However, the authors do not describe how the qPCR method was validated conveniently. In the material and methods section the validation process of the methodology should have been mentioned step by step.
There are several molecular biology methods for epidemiological studies of ESBLs, including detection and quantification by qPCR. I don't understand why the authors tried to implement yet another qPCR method.
In my opinion the results presented do not add anything new to what has been published previously by other authors in terms of E. coli producing ESBLs.
The authors should explore the results a bit more, for example in terms of clones and high-risk international clones. Furthermore, the characterization of the type of ESLs would be interesting as well. The article would be much more interesting.
Round 2
Reviewer 2 Report
This manuscript presents data about extended spectrum b-lactamases (ESBLs) resistance in Escherichia coli isolates from cloacae from chickens during 2016-2019. Moreover, the authors implemented an qPCR method to detect and quantify ESBLs genes as a tool for molecular epidemiology and surveillance studies of ESBLs genes.
The work is interesting and presents important results in epidemiological terms.
The manuscript has undergone fundamental changes for publication. However, the authors should format the tables.
On line 194 the authors wrote anal swabs, but it should be cloacae swabs.
On line 320 should be Enterobacterales.
